# Factors Associated with Gastrointestinal Symptoms among Rotating Shift Nurses in South Korea: A Cross-Sectional Study

**DOI:** 10.3390/ijerph19169795

**Published:** 2022-08-09

**Authors:** Sun-Kyung Hwang, Yun-Ji Lee, Min-Eun Cho, Bo-Kyoung Kim, Yea-In Yoon

**Affiliations:** 1College of Nursing, Pusan National University, Yangsan 50612, Korea; 2Research Institute of Nursing Science, Pusan National University, Yangsan 50612, Korea; 3Department of Nursing, Graduate School, Pusan National University, Yangsan 50612, Korea; 4Department of Nursing, Pusan National University Yangsan Hospital, Yangsan 50612, Korea

**Keywords:** nurses, rotating shift work, gastrointestinal symptoms, psychological distress, sleep

## Abstract

Rotating shift work places a serious burden on nurses’ physical and psychological health. Gastrointestinal (GI) symptoms are a common complaint among shift workers. This study assessed GI symptoms and identified the associations between dietary habits, psychological status, and sleep quality among rotating shift nurses. Data from 125 female nurses in rotating shifts who worked at two tertiary hospitals in South Korea were collected using a questionnaire that included the Gastrointestinal Symptoms Questionnaire; the Dietary Habit Questionnaire; the Depression, Anxiety, Stress Scale (DASS)-21; and the Pittsburgh Sleep Quality Index (PSQI). All participants experienced various GI symptoms, and 47% of them complained of at least one severe GI symptom. There were significant differences in GI symptom scores according to the status of depression, anxiety, stress, and sleep quality. In multiple linear regression analysis, the factors associated with an increase in the occurrence and severity of GI symptoms were poor sleep quality and morbid anxiety and stress. The model explained power at 43.2%. As most nurses in rotating shifts experience GI symptoms, they should receive counseling and training programs at work to alleviate psychological symptoms, improve sleep quality, and pay more attention to their health status as well as GI symptom management.

## 1. Introduction

Shift work, defined as an irregular and unusual work schedule outside the day working times [1,2,3], is inevitable for nurses to provide patient care across 24 h of the day [4]. While shift work effectively maintains continuity of care and service, it disrupts the circadian rhythm and alters hormone secretion through irregular sleep-wake cycles [1,5,6,7]. This may lead to homeostatic imbalance and threaten shift work nurses’ overall health and well-being [1,6,8,9,10,11]. In particular, the health problems related to shift work that nurses experience include inadequate sleep; digestive, metabolic, and psychological disorders; cardiovascular diseases; and cancer [1,3,6,7,12,13].

Gastrointestinal (GI) symptoms and dysfunction, including diarrhea, flatulence, constipation, bloating, and abdominal pain, most commonly occur among shiftwork nurses [2,14,15,16,17,18]. These are often linked to functional bowel disorders (FBDs) or irritable bowel syndrome (IBS) [19]. Although GI symptoms are ignored because they do not cause organic damage [20], severe GI symptoms decrease concentration, increase irritability, and negatively affect the quality of life, work efficiency, and absenteeism rate [15,18,21,22]. Therefore, it is necessary to identify GI symptoms and their associated factors in shiftwork nurses and consider ways to reduce GI problems.

Shift workers experience changes in meal patterns, including skipping meals, consumption at unconventional times, and increased high-fat intake and caffeine consumption [2,7,14]. These habits result in alterations in GI activity and function, which leads to increased GI problems [7,9,14]. In addition, circadian misalignment and sleep deprivation caused by continuously rotating shift work adversely affects the nervous system, which leads to psychological disorders, including depression, anxiety, and stress [7,11,19,23]. These are considered to be associated with GI symptoms due to alterations in cortisol hormone levels [5,24]. Aside from this, shiftwork nurses experience more sleep deprivation and poorer sleep quality following a decline in their ability to maintain sleep homeostasis [4,25,26,27], and their sleep disturbance causes GI symptoms and dysfunction because it results in altered visceral sensitivity and GI motility [7,28,29]. In other words, GI symptoms in shiftwork nurses occur because of the effects of organic links among various factors. Nevertheless, there remains a lack of research on the integrated effect of various variables affecting GI symptoms.

The shift system of Korean nurses generally includes day, evening, and night shifts and is based on a rapidly rotating shift [30], while nurses in other countries have various shift systems, including night shifts, regular rotating shifts, and morning shifts [25,31]. Moreover, at over 80% [30,32], the proportion of rotating shift nurses in Korea is higher than that in other countries [25,33]. Since rapid and irregular rotating shifts create confusion in the awake–sleep cycle, many Korean nurses are likely to suffer health problems such as GI symptoms related to changes in lifestyle, sleep, and psychological status. However, researchers and healthcare managers rarely understand GI problems and methods to improve them among Korean nurses. Therefore, this study aimed to identify the influence of dietary habits, psychological problems, and sleep quality on the GI symptoms of shift work nurses in South Korea and to provide preliminary information to search for a plan to reduce GI symptoms in shift nurses.

## 2. Materials and Methods

### 2.1. Study Design and Participants

This study was a cross-sectional and secondary analysis using part of a study conducted to identify the health problems of hospital nurses in South Korea in 2020 [13]. A total of 167 nurses were enrolled in two tertiary hospitals for a preliminary study. To investigate the general health conditions of nurses, the following exclusion criteria were applied: acute illness or severe illness, surgery or a procedure to treat the illness within three months, and pregnancy [13]. In this study, we defined rotating shift nurses as those who worked in three shifts, including day, evening, and night, for more than one year and were still working in shifts. In addition, we restricted the participants to females to reduce the bias caused by the difference in hormone secretion between males and females, because estrogen, progesterone, and prostaglandins are related to GI symptoms, motility, and sensitivity [34]. We extracted the data of 125 female nurses in rotating shifts from the primary data.

The appropriate sample size for this study was 109, calculated using the G-power program for multiple linear regression with a priori sample size estimation. We computed it as estimated to have a medium effect size of 0.15, significance level of 0.05, and power of 0.80, with eight predictive variables based on previous studies [2,4,15,18], including alcohol consumption, caffeine consumption, healthy dietary habits, unhealthy dietary habits, depression, anxiety, stress, and sleep quality. Therefore, the sample size of this study was sufficient to ensure the reliability and power of the results.

### 2.2. Measurements

#### 2.2.1. Demographic and Job-Related Characteristics

The demographic characteristics examined were age, marital status, living situation, regular exercise, and body mass index (kg/m^2^). Job-related characteristics were the department of work, work experience, and the average workdays of day, evening, and night shifts per month.

#### 2.2.2. Gastrointestinal Symptoms

We used the Gastrointestinal Symptoms Questionnaire [35], which consists of queries on experiencing 16 GI symptoms during the last four weeks, such as abdominal pain, epigastric pain, heartburn, regurgitation, abdominal rumbling, and bloating. Each item is rated on a scale of 0 to 6 (none, mild, moderate, quite a lot, severe, very severe, or unbearable). The total score range is 0–186, and a higher score indicates that the participant complained of more varied and severe symptoms. We categorized the mild and severe symptom groups to analyze the severity of the GI symptoms reported by the participants. These aspects were determined as “none,” “mild,” and “moderate” symptoms for the mild GI symptoms and from “quite a lot” to “unbearable” symptoms for the severe GI symptoms.

#### 2.2.3. Dietary Habits

To identify healthy and unhealthy dietary habits, we analyzed the frequency of food intake using the Dietary Habit Questionnaire [36]. We measured healthy dietary habits as the sum of item scores from a rating of “5” for daily intake, “4” for 5–6 days per week, “3” for 3–4 days, “2” for 1–2 days, and “1” for rare intake, for grains, proteins, fruits, vegetables, and dairy foods. In addition, for each unhealthy item, the sum of item scores for fatty, instant, fast foods, and night meals was computed on a “5” score for over 5 days per week, “4” for 3–4 days, “3” for 1–2 days, “2” for 1–3 days per month, and “1” for rarely intake. Additionally, we investigated alcohol consumption per week and caffeine consumption (such as coffee, tea, and energy drinks) per day.

#### 2.2.4. Psychological Distress

Psychological distress was examined using the Korean version of the Depression, Anxiety, Stress Scale (DASS)-21 downloaded from the DASS Website [37]. The three subscales of depression, anxiety, and stress each had seven items, each of which were assessed using a four-point Likert scale ranging from 0 to 3. The total scores of each seven-item subscale were computed by doubling the sum of the item scores. Each subscale had a cut-off score for conventional severity labels. Cronbach’s alpha for the DASS-21 was 0.94 for depression, 0.87 for anxiety, and 0.91 for stress in the inventory development study, while those in this study were 0.88, 0.79, and 0.87, respectively.

#### 2.2.5. Sleep Quality

Sleep quality was measured with the Korean version of the Pittsburgh Sleep Quality Index (PSQI-K) [38,39]. The global score of the 19 items of the PSQI-K is the sum of seven component scores using a scale from 0 (no difficulty) to 3 (severe difficulty): sleep quality, sleep latency, sleep duration, habitual sleep efficiency, sleep disturbances, use of sleep medication, and daytime dysfunction. A score of 5 or higher was defined as poor sleep quality, and a higher score indicated poorer sleep quality [39]. The reliability coefficient Cronbach’s alpha was 0.83 at development time [38], 0.84 for the PSQI-K [39], and 0.73 in this study.

### 2.3. Statistical Analysis

Data were analyzed using SPSS version 25.0 (IBM, Armonk, NY, USA), with a 0.05 level of significance and two-sided *p*-values. Demographic and job-related characteristics and outcome variables were summarized using descriptive statistics with frequencies, proportions, means, and standard deviations (SD). We compared differences in the total GI symptom score by categorizing the outcome variables using independent t-tests and one-way ANOVA, and the Bonferroni test was used for post hoc analysis. Variables that were not normally distributed or had a sample size of a group lower than 15 were used to conduct non-parametric tests, including the Mann–Whitney or Kruskal–Wallis test. We used the independent *t*-test, chi-squared (χ^2^) test, and Fisher’s exact test to compare the outcome variables between the two groups with and without at least one severe GI symptom. Multiple linear regression was performed to identify variables associated with the severity of GI symptoms in the participants.

## 3. Results

### 3.1. Demographic and Job-Related Characteristics

The mean age of the 125 female rotating shift nurses was 28.0 years (SD = 3.1), and most participants were under 30 years old (*n* = 102, 81.6%). Of the participants, 78.4% were single and 37.6% lived alone. Approximately half of the nurses had regular exercise, and the average BMI was 21.2 kg/m^2^ (SD = 2.8). The largest number of participants worked in the general ward (*n* = 79, 63.2%), followed by intensive care unit nurses (*n* = 29, 23.2%), and emergency room nurses (*n* = 17, 13.6%). The mean work experience was 64.5 months (SD = 35.1), and most nurses (*n* = 118, 94.4%) had worked for under 120 months. The average workdays per month were 6.3 days (SD = 2.1) for day shift, 6.6 days (SD = 2.0) for evening shift, and 5.7 days (SD = 1.2) for night shift (Table 1).

### 3.2. Gastrointestinal Symptoms in Nurses

All participants experienced at least one GI symptom, and 47.2% (*n* = 59) complained of at least one severe symptom. Figure 1 shows the ten GI symptoms frequently reported by the participants. Empty feelings (*n* = 102, 81.6%) were the most prevalent, followed by abdominal rumbling (*n* = 98, 78.4%), bloating (*n* = 96, 76.8%), heartburn (*n* = 83, 66.4%), and postprandial fullness (*n* = 76, 60.8%). The main GI symptoms that nurses complained of were empty feelings (*n* = 26, 20.8%), bloating (*n* = 23, 18.4%), abdominal rumbling (*n* = 20, 16.0%), and postprandial fullness (*n* = 20, 16.0%).

### 3.3. Dietary Habits According to the Severity of GI Symptoms

The mean total score of GI symptoms was 16.4 ± 13.1 (min–max = 1–66). There were no significant differences, although the total score of GI symptoms was higher in the alcohol consumption group, who consumed alcohol more than three times per week (17.2 ± 11.2 vs. 16.4 ± 13.3 scores, *p* = 0.671), and caffeine consumption group, who consumed caffeine more than one cup per day (20.0 ± 13.6 vs. 15.1 ± 12.8 scores, *p* = 0.066) than those in the other groups. When the differences in dietary habits between nurses with (*n* = 59) and without (*n* = 66) at least one severe GI symptom were compared, the mean caffeine consumption per day was significantly higher in the group with severe GI symptoms (0.8 ± 0.5 vs. 0.6 ± 0.4 cup/day, *p* = 0.030). In addition, nurses with severe GI symptoms consumed less healthy foods (*p* = 0.017). In particular, there were significant differences in the frequency of protein (*p* = 0.020) and vegetable (*p* = 0.020) intake between the two groups (Table 2).

### 3.4. Comparison of Psychological Distress and Sleep Quality

As shown in Table 3, the higher the level of depression (*p* < 0.001), anxiety (*p* < 0.001), and stress (*p* < 0.001), the higher the GI symptom score. In addition, the mean GI symptoms total score was significantly higher in the poor sleep quality group than that in the good sleep quality group (17.8 ± 13.4 vs. 8.9 ± 7.8, *p* < 0.001). In comparison with severe symptom and without severe symptom groups, the mean scores for depression (16.2 ± 9.2 vs. 7.8 ± 7.2, *p* < 0.001), anxiety (11.6 ± 10.1 vs. 4.9 ± 5.4, *p* < 0.001), and stress (10.3 ± 9.0 vs. 4.2 ± 4.9, *p* < 0.001) were significantly higher in nurses with severe GI symptoms. The prevalence of mild or moderate depression (74.6% vs. 30.3%), anxiety (62.7% vs. 22.7%), and stress (28.8% vs. 6.0%) was higher in the group with severe symptoms. The sleep quality of nurses with severe GI symptoms was significantly poorer than that of the other group (9.4 ± 3.9 vs. 7.7 ± 3.1, *p* = 0.026), while the proportion of nurses with poor sleep was not different between the two groups (89.8% vs. 80.3%, *p* = 0.218).

### 3.5. Factors Associated with GI Symptoms Score in Nurses

To identify variables that are likely to affect the severity of GI symptoms, we entered depression, anxiety, stress, and sleep quality, which were heterogeneous variables in the GI symptom total score between the groups, as predictive factors, and performed multiple linear regression analyses. Healthy (r = −0.153, *p* = 0.088) and unhealthy dietary habits (r = 0.12, *p* = 0.20) were excluded as predictive factors because they were not correlated with the GI symptom score in the Pearson’s correlation coefficient analysis. We categorized the depression, anxiety, and stress variables as the normal, mild and moderate groups to determine whether psychological distress that occurred was more important than a numerical score of psychological status, and then converted them to dummy variables. Factors associated with an increase in GI symptom total score were the prevalence of anxiety (β = 0.21, *p* = 0.032), stress (β = 0.17, *p* = 0.039), and sleep quality (β = 0.36, *p* < 0.001). The most influential variable was sleep quality; poor sleep quality increased the severity of GI symptoms. The predictive model explained 43.2% of the power (F = 14.54, *p* < 0.001) (Table 4).

## 4. Discussion

In South Korea, many nurses experience rotating shift work over an extended period [30,32,40] and are exposed to common health hazards related to shift work [2,14,16,17]. We examined the prevalence of GI symptoms and the association between GI symptoms and dietary habits, psychological distress and sleep quality to provide preliminary information on managing GI problems among rotating shift nurses in Korea. We found that GI symptoms were significantly common among rotating shift nurses. In addition, the severity of GI symptoms were significantly associated with psychological distress and sleep quality.

In this study, all rotating shift nurses had at least one GI symptom; 47% complained of more than one severe GI symptom, and the most frequent GI symptoms were an empty feeling, abdominal rumbling, bloating, heartburn, and postprandial fullness. Prior research indicated that the prevalence of GI disorders was 72.3% among Egyptian shiftwork nurses, and the most frequent symptom was reflux syndrome [21]. Among Iranian shiftwork nurses [41], 81.9% had at least one GI symptom, and regurgitation and bloating were significant. Although it is difficult to compare the results of these studies owing to the differences in GI symptom measurements, these results indicate that shiftwork nurses suffer from a high incidence of GI problems. Considering that only 30–50% of individuals with persistent GI symptoms visit a hospital [20,42], it is likely that nurses’ GI symptoms are ignored, and their seriousness not recognized. Thus, healthcare professionals and employers should be concerned about GI symptoms among shift nurses and make efforts to detect and manage GI problems.

Our findings showed that nurses with severe GI symptoms consumed more caffeine per day and less protein and vegetables. In other studies [43,44], GI symptoms or FBDs were related to less consumption of carbohydrates, dairy foods, and fruits and more fat and protein consumption. According to a systematic review of the eating habits of shift workers [2], they consumed more caffeine and had unbalanced nutrition. These results are linked to irregular meals, skipping meals, and short mealtimes related to the work environment and conditions [8]. It is necessary to investigate the work environment and cafeteria system because shift workers can find it difficult to make time to eat healthy food due to their constantly changing shifts. A previous study showed that difficulties finding the opportunity to utilize a cafeteria and choose healthy food were the chief barriers to healthier eating habits among shift health workers [45]. Changes in dietary habits due to rotating shift work can adversely affect individuals’ metabolic health as well as GI function [3]. Zoto et al. found that night shift nurses showed a higher risk of diabetes development than hospital day workers [46]. Therefore, employers must build an environment and policies in the workplace to promote balanced and regular healthy dietary intake to reduce GI symptoms among nurses in rotating shifts. In addition, further associations with metabolic disorders need to be explored to understand the health of shift-working nurses.

In this study, the prevalence of depression, anxiety, and stress was 51.2%, 41.6%, and 16.8%, respectively. A higher degree of psychological distress, including depression, anxiety, and stress, was linked to a higher GI symptom score, and the prevalence of psychological distress was significantly higher in nurses in rotating shifts who complained of severe GI symptoms. This is similar to the finding that FBDs are associated with higher levels of depression and anxiety [47]. These results indicate that psychological distress and GI disorders are closely associated. Psychological distress leads to changes in hormone secretion and not only affects GI disorders but also causes various physical problems [5]. Since the prevalence of depression, anxiety, and stress among shift nurses was higher than that among fixed-shift nurses [48,49], greater attention to psychological status and GI symptoms is required for Korean nurses in rotating shifts.

Our finding that GI symptoms are associated with poor sleep quality is similar to previous findings [4,18]. However, in this study, the proportion of nurses with poor sleep quality did not differ from those with and without severe GI symptoms. This may be related to the fact that the participants in this study generally had poor sleep quality (approximately 85%). The shift system for Korean nurses has irregular rotating shifts, rather than diverse patterns [30]. Therefore, many rotating shift nurses experience sleep disorders related to the rapid alteration of the awake–sleep cycle and are more exposed to problems caused by sleep disorders [50,51]. To improve this situation, we should prepare a system and policies to expand the various shift types based on an analysis of the nursing work environment in Korea. Moreover, nurses should be allowed to choose an optimized shift system.

Through multiple linear regression analysis, we found that the severity of GI symptoms was likely to increase among nurses in rotating shifts who had anxiety (β = 0.21) and stress (β = 0.17) problems and had poor sleep quality (β = 0.36). According to a study by Ibrahim et al. [52], the occurrence of irritable bowel syndrome increased 4.3 times in nurses with anxiety and was significantly higher in nurses with depression. In addition, a study of shift nurses in Korea showed that the likelihood of functional dyspepsia was 2.2 times higher in severe stress [15]. Psychological distress worsens GI symptoms by affecting hormone secretion, such as cortisol and serotonin [5], induces visceral hypersensitivity and hyperalgesia, and contributes to GI disorders [29]. Because psychological condition is closely related to GI symptoms, simultaneous efforts are needed to detect and control psychological distress and GI symptoms. Therefore, we must comprehensively enhance physical and mental health through the regular screening of psychological and emotional states.

Poor sleep quality was a crucial predictive factor for worsening GI symptoms in this study and has been shown in previous studies to be an important factor associated with GI problems [4,15,18]. Misalignment of the circadian rhythm caused by shift work affects the GI tract function, including motility, hormone production, gastric acid production, nutrient absorption, and intestinal permeability, and is related to GI symptoms and disorders [9]. Kim et al. showed that the likelihood of IBS was 4.1 times higher and that of functional dyspepsia was 2.3 times higher in poor sleep quality [15]. In the Korean Nurses’ Health Study [30], 88% of Korean nurses worked in rotating shifts, and the mean shift work period was 5.7 years. In this regard, over 70% of Korean nurses experience a decline in sleep quality [53,54]. Accordingly, sleep quality promotion is crucial for decreasing GI symptoms among Korean nurses with a higher proportion of shift work. According to Rizza et al.’s study [55], poor sleep quality was related to higher glycated hemoglobin (HbA1c), and the clock gene ratio, which is a potential indicator of the desynchronization of the master clock, was significantly higher in poor sleepers. Certainly, alterations in the sleep/wake cycle and sleep quality decline increase circadian rhythm disruption, which regulates human biological functions [9] and leads to an increase in physical diseases, including metabolism, psychiatric disorders, lifestyle changes, and stress aggravation [11,16,55]. In other words, it can be said that the foundation of not only GI symptoms but also psychological distress and physical illness in nurses in rotating shifts is the destruction of sleep cycle and deterioration of sleep quality. Since lifestyle (including dietary habits), psychological status, and sleep quality organically affect the GI symptoms of rotating shift nurses, it may be difficult to improve health efficiently by focusing only on modifying specific variables. Thus, it is necessary to examine the characteristics of nurses from various perspectives to establish a system to improve them. In a qualitative study [56], organizational support was the most important factor for nurses’ own healthcare. Therefore, it is possible to improve nurses’ health by reforming eating behaviors, improving stress management, and enhancing sleep hygiene and quality by developing an organizational support system, thus further improving the nursing work environment and promoting the quality and safety of nursing.

The results of this study should be interpreted with caution due to several limitations. First, this study was conducted with rotating shift nurses at only two tertiary hospitals; therefore, it may not be generally representative of nurses in South Korea. Second, as the reading of results relied only on subjective data using questionnaires, there might be a gap between self-reported GI disorders and actual GI problems in nurses. In addition, there was a limitation in that detailed dietary habits such as mealtime, frequency, and quantity of each nutritional intake were not obtained due to the investigation using categorical variables. Finally, the impact of the shift work environment could not be confirmed because this study focused on identifying an association between dietary habits, psychological distress, and sleep quality with GI symptoms. Despite these limitations, this study is significant and suggests a need for efforts to reduce the prevalence of high GI symptoms among nurses. Moreover, it presents management directions and emphasizes the importance of establishing an organizational system for GI symptoms and health promotion by comprehensively confirming the influence of psychological distress and sleep quality on GI symptoms.

## 5. Conclusions

We conducted a descriptive cross-sectional study in two tertiary hospitals in South Korea to identify the association between GI symptoms, eating habits, psychological problems, and sleep quality among nurses in rotating shifts. Many nurses working rotating shifts experience severe GI symptoms. They are influenced by comprehensive variables, such as lifestyle changes and a decline in the homeostasis regulation of physical, psychological, and sleep status caused by circadian rhythm misalignment. In particular, this study showed that the crucial factors related to the severity of GI symptoms are poor sleep quality, anxiety, and stress. We found that GI symptoms, which are often ignored for being are too familiar, must be a principal health problem that demands concern among nurses in rotating shifts. It is likely that these could be improved through the management of sleep and psychological conditions. To improve the health status of nurses in rotating shifts, focus should be placed on continuously identifying the factors that influence GI symptoms and developing a comprehensive program based on changes in lifestyle, psychological status, and sleep quality. Additionally, provisions and policies must be established to ensure a better workplace environment for nurses.

## Figures and Tables

**Figure 1 ijerph-19-09795-f001:**
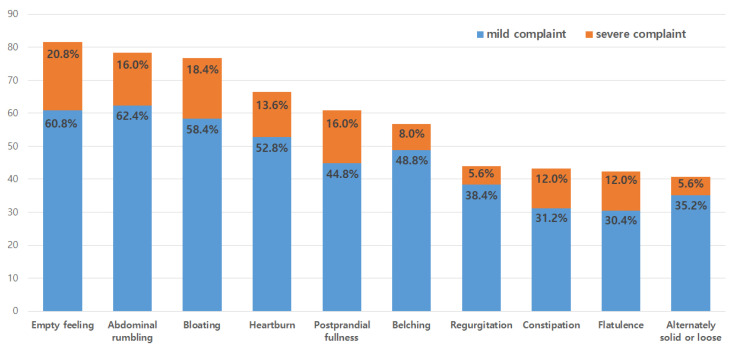
The major gastrointestinal symptoms of participants (*n* = 125).

**Table 1 ijerph-19-09795-t001:** The characteristics of participants (*n* = 125).

Variables	Categories	*n* (%)	M ± SD (Min–Max)
Age	≤25	32 (25.6)	28.0 ± 3.1 (23–38)
(Year)	26–30	70 (56.0)	
	31–35	20 (16.0)	
	≥36	3 (2.4)	
Marital status	Single	98 (78.4)	
	Married	27 (21.6)	
Living with	Alone	47 (37.6)	
	With spouse or children	27 (21.6)	
	With parents	51 (40.8)	
Regular exercise	Yes	55 (44.0)	
	No	70 (56.0)	
BMI	≤22.9	96 (76.8)	21.2 ± 2.8
(Kg/m^2^)	23–24.9	17 (13.6)	
	≥25	12 (9.6)	
Department	General ward	79 (63.2)	
	Intensive care unit	29 (23.2)	
	Emergency room	17 (13.6)	
Nursing career	≤60	56 (44.8)	64.5 ± 35.1 (15–195)
(month)	61–120	62 (49.6)	
	121–180	5 (4.0)	
	≥181	2 (1.6)	
Workdays/month	Day		6.3 ± 2.0 (2–15)
	Evening		6.6 ± 2.0 (1–11)
	Night		5.7 ± 1.2 (2–8)

BMI = body mass index.

**Table 2 ijerph-19-09795-t002:** Dietary habits in accordance with the severity of GI symptoms in participants (*n* = 125).

Variables	Overall (*n* = 125)	With Severe GI Symptoms (*n* = 59)	Without Severe GI Symptoms (*n* = 66)	t or χ^2^ (*p*)
	*n* (%) or M ± SD			
Alcohol consumption				
<three times/wk	116 (92.8)	53 (89.8)	63 (95.5)	1.48 (0.225)
≥three times/wk	9 (7.2)	6 (10.2)	3 (4.5)	
Caffeine consumption	0.7 ± 0.5	0.8 ± 0.5	0.6 ± 0.4	−2.20 (0.030)
<one cup/day	91 (72.8)	38 (64.4)	53 (80.3)	3.98 (0.046)
≥one cup/day	34 (27.2)	21 (35.6)	13 (19.7)	
Healthy dietary habit	17.6 ± 4.5	16.6 ± 4.5	18.5 ± 4.3	2.41 (0.017)
Grains	4.6 ± 0.9	4.5 ± 1.0	4.7 ± 0.7	1.86 (0.066)
Proteins	3.7 ± 1.4	3.7 ± 1.5	3.9 ± 1.3	2.35 (0.020)
Fruits	2.9 ± 1.4	2.7 ± 1.5	3.1 ± 1.3	1.69 (0.094)
Vegetables	3.3 ± 1.4	3.0 ± 1.3	3.6 ± 1.4	2.11 (0.036)
Dairy food	3.1 ± 1.3	3.1 ± 1.3	3.2 ± 1.4	0.35 (0.729)
Unhealthy dietary habit	12.4 ± 2.7	12.6 ± 2.9	12.2 ± 2.4	−0.80 (0.424)
Fatty food	3.3 ± 0.8	3.3 ± 0.8	3.3 ± 0.8	−0.35 (0.725)
Instant food	3.3 ± 0.9	3.3 ± 1.0	3.3 ± 0.9	−0.49 (0.625)
Fast food	2.7 ± 0.8	2.8 ± 0.8	2.7 ± 0.7	−0.26 (0.796)
Night meal	3.0 ± 0.9	3.2 ± 1.0	2.9 ± 0.8	−1.31 (0.193)

**Table 3 ijerph-19-09795-t003:** Psychological status and sleep quality of participants according to the severity of GI symptoms (*n* = 125).

Variables (Score)	Overall *n* (%) or M ± SD	Total Score of GI Symptoms	t or F (*p*)	With Severe GI Symptoms (*n* = 59)	Without Severe GI Symptoms (*n* = 66)	t or χ^2^ (*p*)
		M ± SD (MR)		*n* (%) or M ± SD		
Depression	11.8 ± 9.2			16.2 ± 9.2	7.8 ± 7.2	−5.69 (<0.001)
Normal ^a^ (≤9)	61 (48.8)	9.9 ± 7.8	25.25 (<0.001)	15 (25.4)	46 (69.7)	25.78 (<0.001)
Mild ^b^ (10–13)	16 (12.8)	15.4 ± 8.7	a, b < c *	9 (15.3)	7 (10.6)	
Moderate ^c^ (≥14)	48 (38.4)	25.1 ± 14.8		35 (59.3)	13 (19.7)	
Anxiety	8.1 ± 8.6			11.6 ± 10.1	4.9 ± 5.4	−4.75 (<0.001)
Normal (≤7)	73 (58.4)	10.8 ± 8.9 (47.0)	37.89 (<0.001) ^†^	22 (37.3)	51 (77.3)	23.30 (<0.001)
Mild (8–9)	12 (9.6)	16.8 ± 10.8 (67.7)		6 (10.2)	6 (9.1)	
Moderate (≥10)	40 (32.0)	26.6 ± 14.2 (90.7)		31 (52.5)	9 (13.6)	
Stress	7.1 ± 7.7			10.3 ± 9.0	4.2 ± 4.9	−4.77 (<0.001)
Normal (≤14)	104 (83.2)	13.8 ± 11.0 (56.0)	23.39 (<0.001) ^†^	42 (71.2)	62 (93.9)	13.30 (0.001)
Mild (15–18)	8 (6.4)	25.1 ± 12.0 (92.1)		5 (8.5)	3 (4.5)	
Moderate (≥19)	13 (10.4)	32.5 ± 15.9 (101.0)		12 (20.3)	1 (1.5)	
Sleep quality	8.5 ± 3.6			9.4 ± 3.9	7.7 ± 3.1	−2.25 (0.026)
Poor sleep (≥5)	106 (84.8)	17.8 ± 13.4	4.00 (<0.001)	53 (89.8)	53 (80.3)	1.52 (0.218)
Good sleep (<5)	19 (15.2)	8.9 ± 7.8		6 (10.2)	13 (19.7)	

^a^ = The normal group (non-depression); ^b^ = The mild depression group; ^c^ = The moderate depression group; * = Post-hoc analysis; ^†^ = Kruskal–Wallis test; MR = mean rank.

**Table 4 ijerph-19-09795-t004:** Factors Associated with GI symptoms Score (*n* = 125).

Variables	B	SE	β	t	*p*	95% CI	
						LL	UL
(Constant)	−0.03	2.32		−0.01	0.989		
Depression (ref: normal)	4.30	2.42	0.17	1.78	0.078	−0.49	9.09
Anxiety (ref: normal)	5.45	2.51	0.21	2.17	0.032	0.48	10.42
Stress (ref: normal)	5.85	2.80	0.17	2.09	0.039	0.30	11.40
Sleep quality	1.30	0.27	0.36	4.85	<0.001	0.77	1.83

SE = standard error; CI = confidence interval; LL = lower limit; UL = upper limit.

## Data Availability

Data are available upon request due to restrictions on privacy and ethics. The data presented in this study are available upon request from the corresponding authors.

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
