# Peer review of "Factors Associated with Gastrointestinal Symptoms among Rotating Shift Nurses in South Korea: A Cross-Sectional Study"

_ijerph, 2022, doi:10.3390/ijerph19169795_

Round 1

Reviewer 1 Report

This paper explores the incidence and influencing factors of gastrointestinal symptoms among among rotating shift nurses, and finds that the incidence of gastrointestinal symptoms in this group is very high and points out that anxiety, stress, and sleep quality are significantly correlated with this. Although it is only a single-center cross-sectional survey, I believes that the findings are still very valuable. The author has made appropriate revisions to the article based on my last review comments. I have no further comments for revision.

Reviewer 2 Report

no more concerns

Reviewer 3 Report

Dear authors, 

All suggestions made have been addressed. Therefore, I congratulate the authors for the work. 

This manuscript is a resubmission of an earlier submission. The following is a list of the peer review reports and author responses from that submission.

Round 1

Reviewer 1 Report

This study investigated the incidence of gastrointestinal symptoms and associated risk factors in rotating shift nurses and found that caffeine intake and anxiety and depression were most significantly associated with gastrointestinal symptoms. I think it's a very novel and interesting study, and the results are very valuable for hospital management to take measures to improve the health status of night shift nurses. The design method of the study is sound, the results are presented clearly, and the conclusions are reasonable and well-founded. The only question I have is that many factors were included in the multivariate logistic regression analysis conducted in this article, but the sample size of this study is less than 200 people, whether such a small sample size can satisfy the multivariate logistic regression analysis?Are the statistics reliable?

Author Response

[Comments from Reviewer 1]

Comments

Responses

This study investigated the incidence of gastrointestinal symptoms and associated risk factors in rotating shift nurses and found that caffeine intake and anxiety and depression were most significantly associated with gastrointestinal symptoms. I think it's a very novel and interesting study, and the results are very valuable for hospital management to take measures to improve the health status of night shift nurses. The design method of the study is sound, the results are presented clearly, and the conclusions are reasonable and well-founded.

We would like to thank you for taking the time and effort to review our manuscript. We have rechecked our manuscript based on your comments.

The only question I have is that many factors were included in the multivariate logistic regression analysis conducted in this article, but the sample size of this study is less than 200 people, whether such a small sample size can satisfy the multivariate logistic regression analysis? Are the statistics reliable?

To identify statistics reliability and power of results, we asked for a consultation with the statistician. Thus, we became aware that the populations of our study were not enough, and the findings had a statistical limitation, like your comments. As mentioned in the Methods section, our study was a secondary analysis with extracted 125 nurses from primary data. Since we calculated the sample size of the primary data based on the multiple linear regression, we have not considered the sample size for logistic regression to be used in the secondary analysis. We also considered the statistical method of stepwise entering variables, but we think it can reduce the reliability of the entire research process. Therefore, we added this point to the limitations of the Discussion section without changing the results (lines 301–303 on p.9).

Additionally, we revised the term "univariate and multivariate logistic regression" as "univariable and multivariable logistic regression" according to a statistics consultation and related article (PMID 32215638). (lines 131–132 on p.3; lines 184–185, 189, and Table 4 on p. 6; line 250 on p. 8)

Reviewer 2 Report

In this manuscript Dr Sun-Kyung Hwang and coauthors explored the associations between GI symptoms, dietary habits, psychological status, and sleep quality among 125 south Korean rotating shift nurses. Authors found that, among study covariates, depression symptoms and less intake of healthy diet were associated with severe GI symptoms.

Overall, this is an interesting paper tproviding usefull data regarding a type of symptoms too often understimated or underscored.

However, I have some concerns:

1) It is unclear if the participants are female, male or both. This information must be clear throughout the text, firstly in abstract.

2) While the introduction is very well written, the rest of the manuscript shows some minor syntax and/or grammar errors, please check carefully.

3 Please discuss more extensively the risk of diabetes development among nurses doing night shift (PMID: 30877388)

4) Table 2 and fig. 2 are redundant, please remove fig. 2 (too difficult to interpret)

5) In results, lines 196, rephrase as follows: .., while healthy dietary habits significantly reduced the odds of severe GI symptoms

6) The fact that the poor quality of sleep does not influence the presence of severe GI symptoms deserves a more detailed discussion. In this connection, the unbalanced expression of clock genes due to night shift work may provide an excellent point of discussion (PMID: 33788000)

Author Response

[Comments from Reviewer 2]

Comments

Responses

In this manuscript Dr Sun-Kyung Hwang and coauthors explored the associations between GI symptoms, dietary habits, psychological status, and sleep quality among 125 south Korean rotating shift nurses. Authors found that, among study covariates, depression symptoms and less intake of healthy diet were associated with severe GI symptoms.

Overall, this is an interesting paper providing usefull data regarding a type of symptoms too often understimated or underscored.

We appreciate your considerable review and comments on our work. We have revised the manuscript, focusing on your suggestions.

1) It is unclear if the participants are female, male or both. This information must be clear throughout the text, firstly in abstract.

We included only female rotating shift nurses in our study. Accordingly, to clarify our study’s population, we have revised the participants statement in the abstract to indicate 125 female rotating shift nurses (line 13 on p. 1). Further, a description of why only female nurses’ data were extracted was added in the Methods section (lines 79–82 on p.2). Regarding the primary data, among the overall rotating shift nurses, 125 were female, and 12 were male.

2) While the introduction is very well written, the rest of the manuscript shows some minor syntax and/or grammar errors, please check carefully.

The reviewer’s comment is correct. Thus, we have thoroughly checked our manuscript for errors in syntax, grammar, and typing. In addition, we have availed of the editing service of the editing company Editage (http://www.editage.co.kr) to go over our manuscript, and the manuscript has been examined by a native English speaker. We have attached the English editing confirmation document.

3) Please discuss more extensively the risk of diabetes development among nurses doing night shift (PMID: 30877388)

Thank you very much for your kindness in suggesting relevant articles. We agree with your comment on the need to consider diabetes among night shift nurses. Diabetes is indeed a common and crucial health problem related to the alteration of the sleep–wake cycle. However, our study focused on only GI symptoms and their factors. We think it is difficult to infer the relationship between GI symptoms, influencing factors, and diabetes using only our findings, although GI symptoms are closely associated with diabetes. Even though we could not discuss the risk of diabetes in our study, we will try to conduct further research to identify not only GI symptoms but also diabetes, metabolic disorders, and other health problems among rotating shift nurses.

We have revised the Discussion section to clarify the aim of our study. We would greatly appreciate your review of our revised manuscript (p. 7–8).

4) Table 2 and fig. 2 are redundant, please remove fig. 2 (too difficult to interpret)

We agree with your comments and have removed Figure 2 from the manuscript. Hence, we have deleted the end sentence in Section 3.3 (line 165 on p.5).

5) In results, lines 196, rephrase as follows: .., while healthy dietary habits significantly reduced the odds of severe GI symptoms

We appreciate your definitive suggestion on the direction of revision. We have revised the sentence based on your comment as follows: “The odds of experiencing severe GI symptoms were approximately 2.9 times higher in nurses with mild or moderate depression than in those without depression, while healthy dietary habits significantly reduced the odds of severe GI symptoms.” (lines 192–195, p. 6)

6) The fact that the poor quality of sleep does not influence the presence of severe GI symptoms deserves a more detailed discussion. In this connection, the unbalanced expression of clock genes due to night shift work may provide an excellent point of discussion (PMID: 33788000)

Focusing on your comment, we have attempted to provide a more detailed discussion on the reason for there being no association between severe GI symptoms and poorer sleep quality in our study. We have added content on the fact that most nurses had poor sleep quality; in particular, numerous Korean nurses have poor sleep quality. Further, we have also included a sentence describing sleep quality decline as being associated with circadian rhythm misalignment, lifestyle changes, increased psychological stress, and decreased GI tract function (lines 274–286, p. 8).

Reviewer 3 Report

Dear authors,

Thank you for the opportunity to review your interesting and well written manuscript. I will give my feedback following the structure of the manuscript. 

1.The title is informative and the abstract provides a summary of the manuscript's major aspects.

2.The background chapter is well written, providing convincing arguments for the need of the study. A problem statement is well formulated and the aim is relevant.  

3.Results: I recommend the authors to present Figure 2 more clearly, especially the legend, now it’s hard to read. I also recommend the authors to revise Table 3 to provide information about the units of the results. 

4.Discussion chapter is clearly and well written but I suggest the authors check that the information is presented in the correct order according to the objectives as there is information that is being shared throughout the discussion.  Also I recommend the authors to review the sentence form line 233 to 235. Should the references appear in the middle of the sentence or at the end? In lines 264 to 267 appear this sentence: “previous study showed that difficulties finding the opportunity to utilize a cafeteria and to choose healthy food were the chief barriers to healthier eating habits among shift health workers”.  A reference is needed. 

5.Under my point of view conclusions should include the results associated with severe GI symptoms in the multivariate regression analysis. In this section I would also like to read the implications of the study.

Author Response

[Comments from Reviewer 3]

Comments

Responses

Thank you for the opportunity to review your interesting and well written manuscript. I will give my feedback following the structure of the manuscript. 

Thank you very much for taking the precious time to review our manuscript. We appreciate your constructive comments and have incorporated your suggestions into the revised manuscript.

1. The title is informative and the abstract provides a summary of the manuscript's major aspects.

2. The background chapter is well written, providing convincing arguments for the need of the study. A problem statement is well formulated and the aim is relevant.  

Thank you for raising this point. We have rechecked our overall manuscript.

3-1. Results: I recommend the authors to present Figure 2 more clearly, especially the legend, now it’s hard to read.

We confirm that Figure 2 was unclear and redundant with Table 2 according to the reviewer 2’s comment. Thus, we have removed Figure 2 and the associated sentence from the manuscript (line 165 on p.5).

3-2. I also recommend the authors to revise Table 3 to provide information about the units of the results. 

The comment is correct. We have included the units of the statistical results in Tables 2 and 3 (p. 5–6).

4-1. Discussion chapter is clearly and well written but I suggest the authors check that the information is presented in the correct order according to the objectives as there is information that is being shared throughout the discussion.

We have looked at the overall Discussion section and revised the contents to clarify our study’s purpose. Mainly, the contents related to the findings of our multivariable logistic regression have been revised (p. 8).

4-2. Also I recommend the authors to review the sentence form line 233 to 235. Should the references appear in the middle of the sentence or at the end?

We agree with your comment and have included the references at the end of the sentence. Further, we have also revised the sentence for clarity (lines 236–238, p. 7).

4-3. In lines 264 to 267 appear this sentence: “previous study showed that difficulties finding the opportunity to utilize a cafeteria and to choose healthy food were the chief barriers to healthier eating habits among shift health workers”.  A reference is needed. 

The reviewer’s comment is correct. We have added a reference (lines 267–269, p. 8).

5. Under my point of view conclusions should include the results associated with severe GI symptoms in the multivariate regression analysis. In this section I would also like to read the implications of the study.

We have revised our conclusions to summarize the study’s aims and results. Based on your comments, we have included the findings of our multivariable regression analysis in the Conclusions section and presented the implication of our study and direction for further research (lines 317–319 and lines 322–330, p. 9).

Round 2

Reviewer 2 Report

Unfortuntelly, authors did not fully treasure my suggestions. In fact they did not discuss the risk of diabetes among nurses doing night shift, did not provide a detailed discussion explaining the lack of correlation between quality of sleep and severe GI symptoms, did not go into the merits of the very likely relation between clock genes dysregulation and GI symptoms (papers suggested). Basically, they did not give sufficient consideration to my previous revision. 

more detailed discussion